# Dual Manifold Adversarial Robustness: Defense against $L_p$ and non-$L_p$ Adversarial Attacks

**Wei-An Lin**[*][†]
University of Maryland
walin@umd.edu

**Chun Pong Lau**[*]
Johns Hopkins University
clau13@jhu.edu

**Alexander Levine**
University of Maryland
alevine0@cs.umd.edu

**Rama Chellappa**
Johns Hopkins University
rchella4@jhu.edu

**Soheil Feizi**
University of Maryland
sfeizi@cs.umd.edu

## Abstract

Adversarial training is a popular defense strategy against attack threat models with bounded $L_p$ norms. However, it often degrades the model performance on normal images and more importantly, the defense does not generalize well to novel attacks. Given the success of deep generative models such as GANs and VAEs in characterizing (approximately) the underlying manifold of images, we investigate whether or not the aforementioned deficiencies of adversarial training can be remedied by exploiting the underlying manifold information. To partially answer this question, we consider the scenario when the manifold information of the underlying data is available. We use a subset of ImageNet natural images where an approximate underlying manifold is learned using StyleGAN. We also construct an "On-Manifold ImageNet" (OM-ImageNet) dataset by projecting the ImageNet samples onto the learned manifold. For this dataset, the underlying manifold information is exact. Using OM-ImageNet, we first show that adversarial training in the latent space of images (i.e. on-manifold adversarial training) improves both standard accuracy and robustness to on-manifold attacks. However, since no out-of-manifold perturbations are realized, the defense can be broken by $L_p$ adversarial attacks. We further propose *Dual Manifold Adversarial Training (DMAT)* where adversarial perturbations in both latent and image spaces are used in robustifying the model. Our DMAT improves performance on normal images, and achieves comparable robustness to the standard adversarial training against $L_p$ attacks. In addition, we observe that models defended by DMAT achieve improved robustness against novel attacks which manipulate images by global color shifts or various types of image filtering. Interestingly, similar improvements are also achieved when the defended models are tested on (out-of-manifold) natural images. These results demonstrate the potential benefits of using manifold information (exactly or approximately) in enhancing robustness of deep learning models against various types of novel adversarial attacks. Codes and models will be available in this link.

## 1 Introduction

Deep neural networks have achieved impressive success in several fields including computer vision, speech, and robot control [1, 2, 3]. However, they are vulnerable against *adversarial attacks* [4, 5, 6, 7,

---

[*]First two authors contributed equally.

[†]Wei-An Lin is now at Adobe.

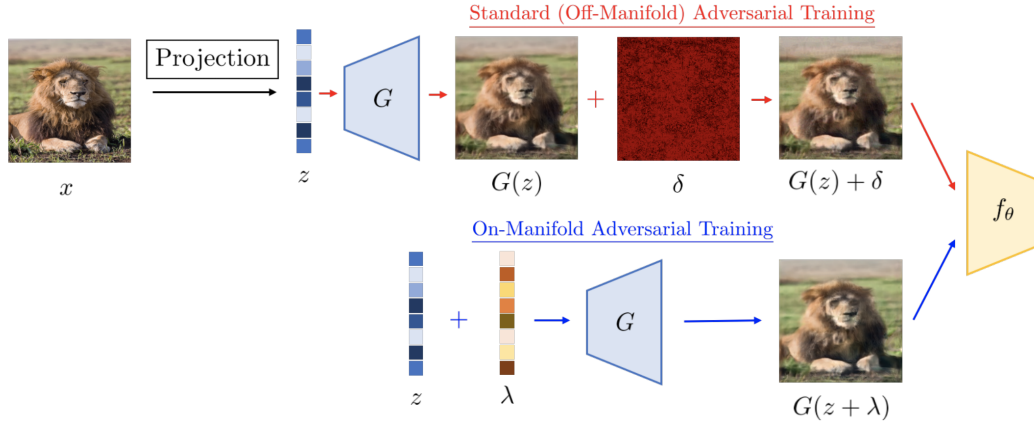

Figure 1: The overall pipeline of the proposed Dual Manifold Adversarial Training (DMAT). In this paper, we consider the scenario when the information about the image manifold is available. This is achieved by projecting natural images $x$ onto the range space of a trained generative model $G$. We empirically show that either standard adversarial training or on-manifold adversarial training alone does not provide sufficient robustness, while DMAT achieves improved robustness against unseen attacks. During test time, images are directly passed to the adversarially trained classifier.

8, 9] which add, often imperceptible, manipulations to inputs to mislead the model. Sensitivity against adversarial attacks poses a huge challenge in security-critical applications where these networks are used. To improve the robustness of deep neural networks against different adversarial threat models, several empirical and certifiable defense methods have been proposed in the past few years including empirical defenses [10, 11, 12, 13], certifiable defenses [14, 15, 16, 17, 18, 19] and defences that detect and reject adversarial examples [20, 21, 22].

Among all of these defenses, *Adversarial training* (AT) [23], which augments the training data with adversarial examples, is perhaps the most standard one. Most existing AT methods consider adversarial distortions within a small $L_p$ ball, and demonstrate robustness to the same type of distortion. However, robustness to $L_p$ distortions often comes with the cost of reduced standard accuracy [24]. Moreover, models trained solely on $L_p$ attacks are shown to generalize poorly to unforeseen attacks and are vulnerable to imperceptible color shifts [25], image filtering [26], and adversarial examples on the data manifold [27, 28]. Being robust against unseen attacks is critical in practice since adversaries will not follow a particular attack threat model.

Existing defenses consider properties of the trained classifier while ignoring the particular structure of the underlying image distribution. Recent advances in GANs and VAEs [29, 30, 31] are shown to be successful in characterizing the underlying image manifold[3]. This motivates us to study whether or not leveraging the underlying manifold information can boost robustness of the model, in particular against unseen and novel adversarial attacks. Our key intuition is that, in many cases, the latent space of GANs and VAEs represent compressed semantic-level features of images. Thus, robustifying in the latent space may guide the classification model to use robust features instead of achieving high accuracy by exploiting non-robust features in the image space [32].

In this paper, we attempt to answer this question by considering the scenario when the manifold information of the underlying data is available. First, we construct an "On-Manifold ImageNet" (OM-ImageNet) dataset where all the samples lie *exactly* on a low-dimensional manifold. This is achieved by first training a StyleGAN on a subset of ImageNet natural images, and then projecting the samples onto the learned manifold. With this dataset, we show that an on-manifold adversarial training (i.e. adversarial training in the latent space) could *not* defend against standard off-manifold attacks and vice versa. This motivates us to propose **D**ual **M**anifold **A**dversarial **T**raining (**DMAT**), which mixes both off-manifold and on-manifold AT (see Figure 1). AT in the image space (i.e.

off-manifold AT) helps improve the robustness of the model against $L_p$ attacks while AT in the latent space (i.e. on-manifold AT) boosts the robustness of the model against unseen non-$L_p$ attacks.

Over the OM-ImageNet, we empirically show that DMAT leads to comparable robustness with standard AT against $L_\infty$ attacks. Moreover, DMAT significantly outperforms standard AT against novel $L_p$ and non-$L_p$ attacks. For example, *for unseen $L_2$ and Fog attacks, DMAT improves the accuracy by 10% compared to standard adversarial training*. Interestingly, similar improvements are also achieved when the defended models are evaluated on (out-of-manifold) natural images. These results shed some light on the use of the underlying manifold information of images to enhance the intrinsic robustness of the models, specially against novel adversarial attacks.

## 2   Preliminaries

**Setup.** We consider the classification task where the image samples $x \in \mathcal{X} := \mathbb{R}^{H \times W \times C}$ are drawn from an underlying distribution $\mathbb{P}_X$. Let $f_\theta$ be a parameterized model which maps any image in $\mathcal{X}$ to a discrete label $y$ in $\mathcal{Y} := \{1, \cdots, |\mathcal{Y}|\}$. An *accurate* classifier maps an image $x$ to its corresponding true label $y_{\text{true}}$, i.e. $f_\theta(x) = y_{\text{true}}$. Without loss of generality, we assume $\mathbb{P}_X$ is supported on a lower-dimensional manifold $\mathcal{M}$ and can be approximated by $\mathbb{P}_{G(Z)}$ where $G(\cdot)$ is a generative model and $Z$ is distributed according to the normal distribution. Informally, we refer to the support of $G(Z)$ as the approximate image manifold $\bar{\mathcal{M}}$ for $\mathcal{M}$. We say the manifold information is *exact*, when there exists a generative model $G$ such that $\mathcal{M} = \bar{\mathcal{M}}$.

**Standard adversarial robustness.** We say $x_{adv}$ is an adversarial example of the input image $x$ if $f_\theta(x) = y_{\text{true}}$, $f_\theta(x_{adv}) \neq y_{\text{true}}$, and the difference between $x_{adv}$ and $x$ are imperceptible[4]. Most existing works consider $L_p$ additive attacks where $x_{adv} = x + \delta$ subject to $\|\delta\|_p < \epsilon$. Formally,

$$\max_{\delta \in \Delta} \mathcal{L}(f_\theta(x + \delta), y_{\text{true}}), \tag{1}$$

where $\Delta = \{\delta : \|\delta\|_p < \epsilon\}$ and $\mathcal{L}$ is a classification loss function (e.g. the cross-entropy loss). In this paper, we focus on $p = \infty$, and will explicitly specify the type of norm when $p \neq \infty$ is used. In (1), since the function is non-convex, the maximization is typically performed using gradient-based optimization methods. The Fast Gradient Sign Method (FGSM) [5] is an $L_\infty$ attack which uses the sign of the gradient to craft adversarial examples:

$$\delta = \epsilon \cdot sign\left(\nabla_x \mathcal{L}(f_\theta(x), y_{\text{true}})\right). \tag{2}$$

The Projected Gradient Descent (PGD) [23] attack is an iterative version of FGSM, which applies $K$ steps of gradient descent. In the rest of the paper, we will use the notation PGD-$K$ to represent $K$-step PGD attacks with bounded $L_\infty$ norm.

To defend against norm-bounded attacks, an established approach by Madry *et al.* [23] considers the following min-max formulation:

$$\min_\theta \sum_i \max_{\delta \in \Delta} \mathcal{L}(f_\theta(x_i + \delta), y_{\text{true}}), \tag{3}$$

where the classification model $f_\theta$ is trained exclusively on adversarial images by minimizing the cross-entropy loss. This approach is called adversarial training (AT). Notice that when the manifold information is exact, we have $x_i = G(z_i)$ for some $z_i$. Thus, the standard adversarial training can be expressed as:

$$\min_\theta \sum_i \max_{\delta \in \Delta} \mathcal{L}(f_\theta(G(z_i) + \delta), y_{\text{true}}). \tag{4}$$

**On-manifold adversarial robustness.** The concept of on-manifold adversarial examples has been proposed in prior works [33, 27, 34]. For any image $x_i \in \mathcal{M}$, we can find the corresponding sample $G(z_i)$ on $\bar{\mathcal{M}}$ which best approximate $x_i$, where $z_i = \arg\min_z \|G(z) - x_i\|_2$. Adversarial examples that lie on the manifold can then be crafted by manipulating the latent representation $z_i$.

$$\max_{\lambda \in \Lambda} \mathcal{L}(f_\theta(G(z_i + \lambda)), y_{\text{true}}), \tag{5}$$

where $\Lambda = \{\lambda : \|\lambda\|_\infty < \eta\}$. Similar to standard adversarial attacks in the image space, the maximization in (5) can be performed by FGSM or PGD-$K$, which we denote as OM-FGSM

and OM-PGD-$K$ respectively. In [27], it has been shown that deep learning models trained by standard AT (3) can be fooled by on-manifold adversarial examples. To defend against on-manifold attacks, [34] considers a similar mini-max formulation in the latent space:

$$\min_{\theta} \sum_i \max_{\lambda \in \Lambda} \mathcal{L}(f_{\theta}(G(z_i + \lambda)), y_{\text{true}}). \tag{6}$$

This approach is called the on-manifold adversarial training (OM-AT). In [34], on-manifold adversarial training has been shown to improve generalization on datasets including EMNIST [35], Fashion-MNIST [36], and CelebA [37]. In [33], the authors finetune an adversarially trained model with on-manifold adversarial examples and demonstrate improved robustness on MNIST [38]. Results from these works are restricted to artificial images where only limited textures are present. However, it has been shown that the vulnerability of deep learning models actually results from the fact that the models tend to exploit non-robust features in natural images in order to achieve high classification accuracy [32]. Therefore, whether the manifold information can be used to enhance the robustness of deep learning models trained on natural images is underexplored.

**Notations.** To precisely specify adversarial training procedures, in the rest of the paper, we will add the attack threat model that an adversarial training algorithm uses. For example, "AT [PGD-5]" refers to the standard adversarial training algorithm (4) with the PGD-5 ($L_{\infty}$) attack used during training, and "OM-AT [OM-FGSM]" refers to the on-manifold adversarial training algorithm (6) with OM-FGSM ($L_{\infty}$) as the attack method used during training.

## 3    On-Manifold ImageNet

One major difficulty of investigating the potential benefit of manifold information in general cases is the inability to obtain such information exactly. For approximate manifolds, the effect of the distribution shift between $\mathcal{M}$ and $\bar{\mathcal{M}}$ is difficult to quantify, leading to inconclusive evaluations. To address the issue, we propose a novel dataset, called On-Manifold ImageNet (OM-ImageNet), which consists of images that lie exactly on-manifold.

Our OM-ImageNet is build upon the Mixed-10 dataset introduced in the `robustness` library [39], which consists of images from 10 superclasses of ImageNet. We manually select 69,480 image-label pairs as $\mathcal{D}_{tr}^o = \{(x_i, y_i)\}_{i=1}^N$ and another disjoint 7,200 image-label pairs as $\mathcal{D}_{te}^o = \{(x_j, y_j)\}_{j=1}^M$, both with balanced classes. We first train a StyleGAN [40] to characterize the underlying image manifold of $\mathcal{D}_{tr}^o$. Formally, the StyleGAN consists of a mapping function $h : \mathcal{Z} \to \mathcal{W}$ and a synthesis network $\tilde{g} : \mathcal{W} \to \mathcal{X}$. The mapping function takes a latent code $z$ and outputs a style code $w$ in an intermediate latent space $\mathcal{W}$. Then, the synthesis network takes the style code and produces a natural-looking image $\tilde{g}(w)$. We follow [41] and consider the extended latent space of StyleGAN. In [41], it has been shown that embedding images into the extended latent space is easier than $\mathcal{Z}$ or $\mathcal{W}$ space. Therefore, in the following, we consider $g : \mathcal{W}^+ \to \mathcal{X}$ as the generator function which approximates the image manifold for $\mathcal{D}_{tr}^o$.

In order to obtain images that are completely on-manifold, for each image $x_i$ in $\mathcal{D}_{tr}^o$ and $\mathcal{D}_{te}^o$, we project onto the learned manifold by solving for its latent representation $w_i$ [41]. We use a weighted combination of the Learned Perceptual Image Patch Similarity (LPIPS) [42] and $L_1$ loss to measure the closeness between $g(w)$ and $x_i$. LPIPS is shown to be a more suitable measure for perceptual similarity than conventional metrics, and its combination with $L_1$ or $L_2$ loss has been shown successful for inferring the latent representation of GANs. We adopt this strategy and solve for the latent representation by:

$$w_i = \arg\min_w \ \text{LPIPS}(g(w), x_i) + \|g(w) - x_i\|_1. \tag{7}$$

In summary, the resulting on-manifold training and test sets can be represented by: $\mathcal{D}_{tr} = \{(w_i, g(w_i), x_i, y_i)\}_{i=1}^N$, and $\mathcal{D}_{te} = \{(w_j, g(w_j), x_j, y_j)\}_{j=1}^M$, where $N = 69,480$ and $M = 7,200$. The total number of categories is 10. Notice that for OM-ImageNet, the underlying manifold information is exact, which is given by $\{g(w), w \in \mathcal{W}^+\}$. Sample on-manifold images $g(w_i)$ from OM-ImageNet are presented in Figure 2. On-manifold images in OM-ImageNet have diverse textures, object sizes, lightening, and poses, which is suitable for investigating the potential benefits of using manifold information in more general scenarios.

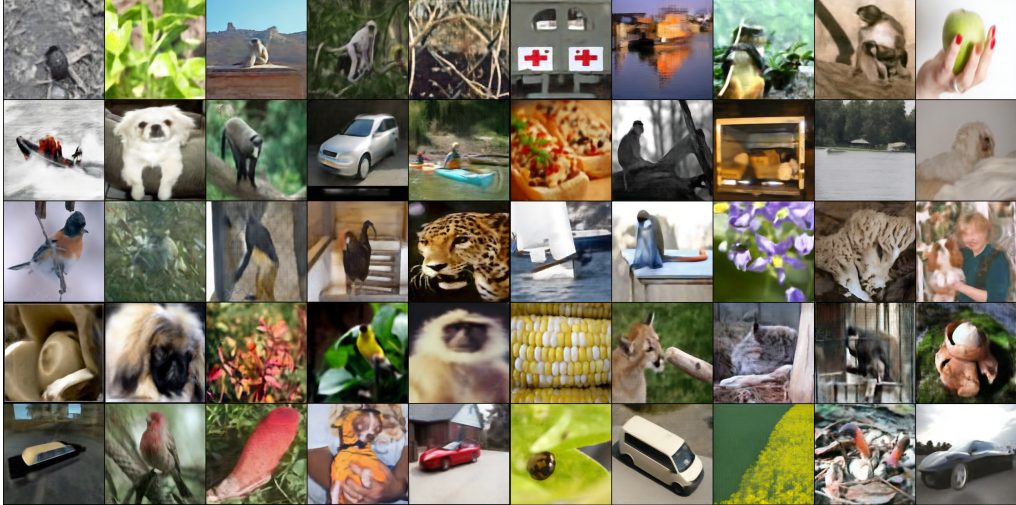

Figure 2: Sample images from the OM-ImageNet dataset. Unlike MNIST-like [38, 35, 36] or CelebA [37] datasets, images in OM-ImageNet have diverse textures. Moreover, the underlying manifold information for this dataset is *exact*.

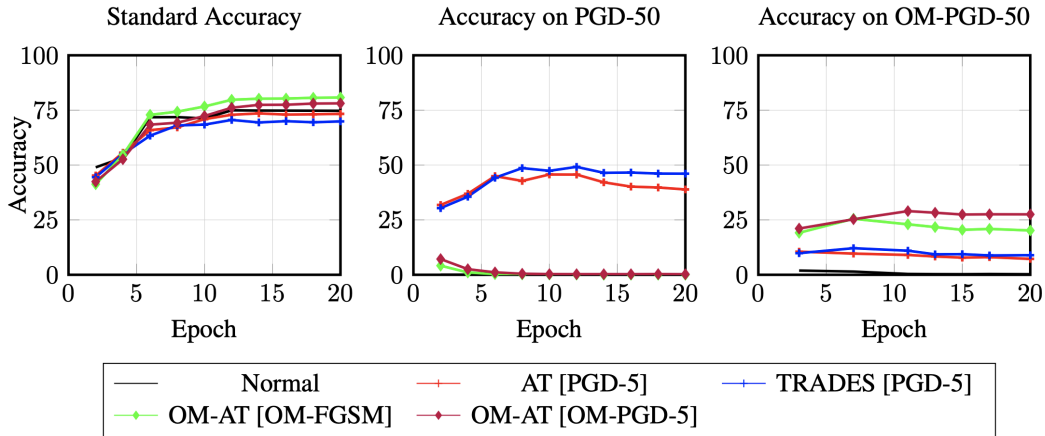

Figure 3: On-manifold adversarial training does not provide robustness to standard attacks. Standard adversarial training does not provide robustness to on-manifold attacks. Left: standard accuracy. Middle: classification accuracy when the trained models are attacked by PGD-50. Right: classification accuracy when the trained models are attacked by OM-PGD-50.

## 4 On-Manifold AT Cannot Defend Standard Attacks and Vice Versa

One issue we would like to investigate first is whether the manifold information alone can improve robustness. We use OM-ImageNet and train several ResNet-50 models to achieve standard $L_\infty$ robustness at a radius of $\epsilon = 4/255$ (according to optimization (8)) or on-manifold robustness at a radius of $\eta = 0.02$ in the latent space (according to optimization (9)).

$$\min_\theta \sum_i \max_\delta \mathcal{L}(f_\theta(g(w_i) + \delta), y_{\text{true}}), \ \ s.t. \ \|\delta\|_\infty < \epsilon. \tag{8}$$

$$\min_\theta \sum_i \max_\lambda \mathcal{L}(f_\theta(g(w_i + \lambda)), y_{\text{true}}), \ \ s.t. \ \|\lambda\|_\infty < \eta. \tag{9}$$

During training, we use the PGD-5 threat model in the image space for (8), whereas for (9) we consider OM-FGSM and OM-PGD-5 as the threat models. For completeness, we also consider robust training using TRADES ($\beta = 6$) [43] in the image space using the PGD-5 threat model. All the

Table 1: Classification accuracy for PGD-50 and OM-PGD-50 attacks on OM-ImageNet test set.

| Method | Standard | FGSM | PGD-50 | MIA | Worst Case | OM-PGD-50 |
|---|---|---|---|---|---|---|
| Normal Training | 74.72% | 2.59% | 0.00% | 0.00% | 0.00% | 0.26% |
| AT [PGD-5] | 73.31% | 48.02% | 38.88% | 39.21% | 38.80% | 7.23% |
| OM-AT [OM-FGSM] | 80.77% | 17.15% | 0.03% | 0.01% | 0.01% | 20.19% |
| OM-AT [OM-PGD-5] | 78.10% | 21.68% | 0.25% | 0.12% | 0.10% | 27.53% |
| DMAT [PGD-5, OM-PGD-5] | 77.96% | 49.12% | 37.86% | 37.65% | 36.66% | 20.53% |

models are trained by the SGD optimizer with the cyclic learning rate scheduling strategy in [44], momentum 0.9, and weight decay $5 \times 10^{-4}$ for a maximum of 20 epochs.

We evaluate the trained models using the PGD-50 and OM-PGD-50 attacks for multiple snapshots during training. The results are presented in Figure 3. We observe that (i) standard adversarial training leads to degraded standard accuracy while on-manifold adversarial training improves it (consistent with [24, 34]) and (ii) standard adversarial training does not provide robustness to on-manifold attacks (consistent with [27]). Interestingly, we also observe that (iii) on-manifold adversarial training does *not* provide robustness to $L_\infty$ attacks since no out-of-manifold samples are realized during training, and (iv) standard adversarial training does not provide robustness to on-manifold attacks.

## 5 Proposed Method: Dual Manifold Adversarial Training

The fact that standard adversarial training and on-manifold adversarial training bring complimentary benefits to the model robustness motivates us to consider the following *dual manifold adversarial training (DMAT)* framework:

$$\min_\theta \sum_i \left\{ \max_{\delta \in \Delta} \mathcal{L}(f_\theta(g(w_i) + \delta), y_{\text{true}}) + \max_{\lambda \in \Lambda} \mathcal{L}(f_\theta(g(w_i + \lambda)), y_{\text{true}}) \right\}, \tag{10}$$

where we are trying to solve for a classifier $f_\theta$ that is robust to both off-manifold perturbations (achieved by the first term) and on-manifold perturbations (achieved by the second term). The perturbation budgets $\Delta$ and $\Lambda$ control the strengths of the threat models during training. For evaluation purposes, we consider PGD-5 and OM-PGD-5 as the standard and on-manifold threat models during training with the same perturbation budgets used by AT and OM-AT. To optimize (10), in each iteration, both standard and on-manifold adversarial examples are generated for the classifier $f_\theta$, and $f_\theta$ is updated by the gradient descent method. The robust model is trained with the identical optimizer setting as in Section 4.

### 5.1 DMAT Improves Generalization and Robustness

Table 1 presents classification accuracies for different adversarial training methods against standard and on-manifold adversarial attacks. For standard adversarial attacks, we consider a set of adaptive attacks: FGSM, PGD-50 and the Momentum Iterative Attack (MIA) [45]. We also report the per-sample worst case accuracy, where each test sample will be viewed as mis-classified if one of the attacks fools the classifier. For on-manifold adversarial attacks, we consider OM-PGD-50. Compared to standard adversarial training, DMAT achieves improved generalization on normal samples, and significant boost for on-manifold robustness, with a slightly degraded robustness against PGD-50 attack (with $L_\infty$ bound of $4/255$). Compared to on-manifold adversarial training (OM-AT [OM-FGSM] and OM-AT [OM-PGD-5]), since out-of-manifold samples are realized for DMAT, robustness against the PGD-50 attack is also significantly improved.

In Figure 4, we visualize PGD-50 and OM-PGD-50 adversarial examples crafted for the trained models. We can see that for the PGD-50 attack, stronger high-frequency perturbations (colored in red) are crafted for models trained by standard adversarial training and DMAT. For the OM-PGD-50 attack, manipulations in semantic-related regions need to be stronger in order to mislead models trained by on-manifold adversarial training and DMAT.

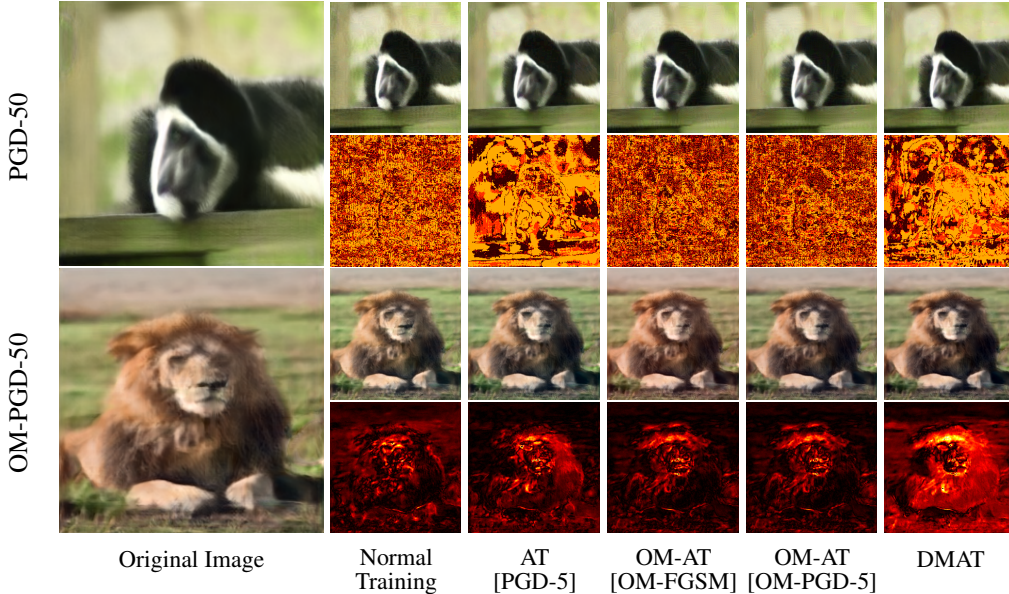

|  |  |  |  |  |  |
|--|--|--|--|--|--|
| Original Image | Normal Training | AT [PGD-5] | OM-AT [OM-FGSM] | OM-AT [OM-PGD-5] | DMAT |

Figure 4: Examples of crafted adversarial examples with the corresponding difference images to the original images using PGD-50 (top) and OM-PGD-50 (bottom) attacks for different robustified models. Brighter colors indicate larger distortions. We can observe that the PGD-50 adversary causes stronger perturbations for classifiers trained with AT and DMAT, and the OM-PGD-50 adversary causes stronger perturbations for the classifier defended by DMAT. In these cases, the model trained by DMAT is more robust and requires stronger distortions to break.

## 5.2 DMAT Improves Robustness to Unseen Attacks

After demonstrating the improved robustness on known attacks brought by DMAT, we investigate whether DMAT improves robustness against novel attacks. We consider several perceptual attacks proposed in [26] including Fog, Snow, Gabor, Elastic, JPEG, $L_2$, and $L_1$ attacks, which apply global color shifts and image filtering to the normal images. Notice that we do not consider the extremely high attack strengths as in [26] since we focus on untargeted attacks. Results presented in Table 2 demonstrate that compared to standard adversarial training, DMAT is more robust against these attacks that are not seen during training. In Figure 5, we visualize the Fog, JPEG compression, and Snow adversarial examples for the trained models. These novel adversaries need to cause stronger distortions on DMAT than standard adversarial training and on-manifold adversarial training to mislead the corresponding classifiers.

Table 2: Classification accuracy against unseen attacks applied to OM-ImageNet test set.

| Method | Fog | Snow | Elastic | Gabor | JPEG | $L_2$ | $L_1$ |
|--------|-----|------|---------|-------|------|-------|-------|
| Normal Training | 0.03% | 0.06% | 1.20% | 0.03% | 0.00% | 1.70% | 0.00% |
| AT [PGD-5] | 19.76% | 46.39% | 50.32% | 50.43% | 10.23% | 41.98% | 21.21% |
| OM-AT [OM-FGSM] | 11.12% | 13.82% | 34.07% | 1.50% | 0.26% | 2.27% | 8.59% |
| OM-AT [OM-PGD-5] | 22.39% | 28.38% | 48.74% | 5.19% | 0.49% | 5.92% | 14.67% |
| DMAT [PGD-5, OM-PGD-5] (**Ours**) | **31.78%** | **51.19%** | **56.09%** | **51.61%** | **14.31%** | **51.36%** | **29.68%** |

## 5.3 TRADES for DMAT

The proposed DMAT framework is general and can be extended to other adversarial training approaches such as TRADES [43]. TRADES is one of the state-of-the-art methods that achieves better trade-off between standard accuracy and robustness compared to standard AT (3). We adopt TRADES in DMAT by considering the following loss function:

$$\min_\theta \sum_i \mathcal{L}(f_\theta(x_i), y_{true}) + \beta \max_\delta \mathcal{L}(f_\theta(x_i), f_\theta(x_i + \delta)) + \beta \max_\lambda \mathcal{L}(f_\theta(x_i), f_\theta(g(w_i + \lambda))), \quad (11)$$

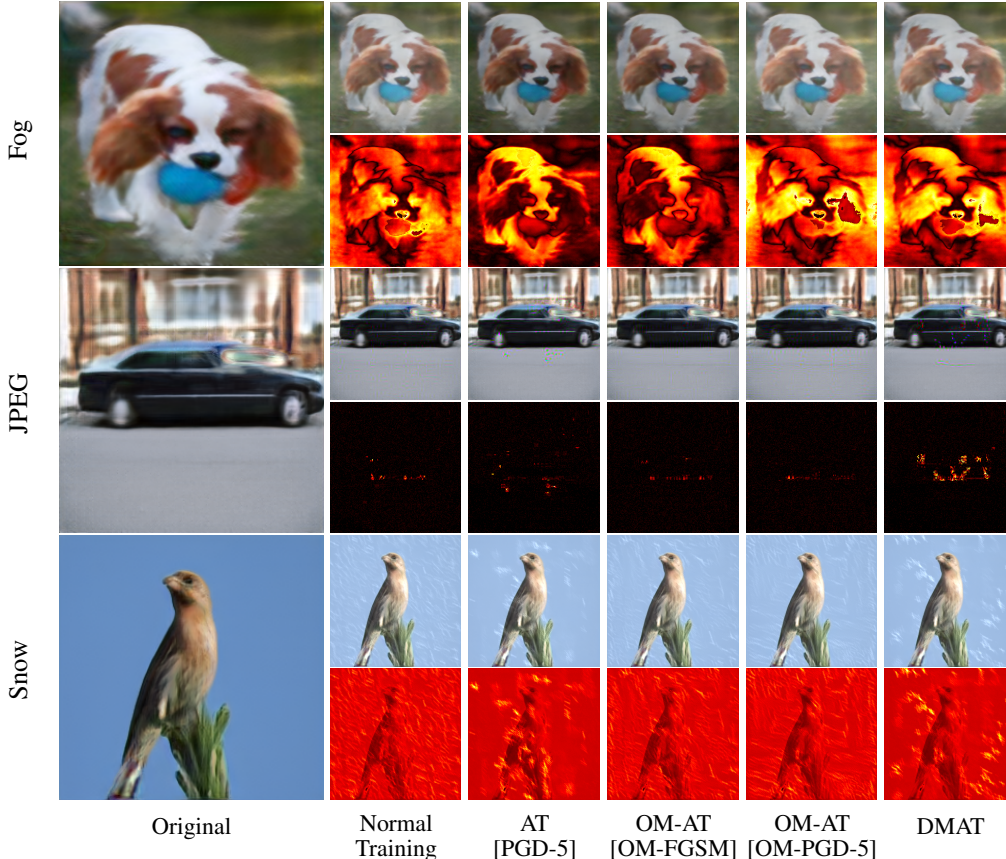

Figure 5: Examples of crafted adversarial examples using the Fog attack (top row), the JPEG compression attack (middle row) and the snow attack (bottom row) for different robustified models. Brighter colors indicate larger absolute differences. We can observe that the classifier trained with DMAT is more robust and needs stronger distortions to break.

where $x_i = g(w_i)$. The first two terms in (11) are the original TRADES in the image space, and the third term is the counterpart in the latent space. To solve for the two maximization problems in (11), we use PGD-5 and OM-PGD-5 with the same parameter settings. Results are presented in Table 3.

Table 3: Classification accuracy against known (PGD-50 and OM-PGD-50) and unseen attacks applied to OM-ImageNet test set. Even for TRADES, the benefit of using manifold information can also be observed.

| Method | Standard | PGD-50 | OM-PGD-50 | Fog | Snow | Elastic | Gabor | JPEG | $L_2$ |
|---|---|---|---|---|---|---|---|---|---|
| Normal Training | 74.72% | 0.00% | 0.26% | 0.03% | 0.06% | 1.20% | 0.03% | 0.00% | 1.70% |
| TRADES | 69.88% | **46.06%** | 8.92% | 18.14% | **47.63%** | 53.32% | **54.33%** | 14.06% | 46.36% |
| DMAT + TRADES | 73.17% | 42.57% | **26.82%** | **30.64%** | 46.62% | **56.38%** | 53.43% | **23.62%** | **55.09%** |

## 5.4 Evaluations on Out-of-Manifold (Natural) Samples

In previous sections, we have considered adversaries crafting adversarial perturbations and applying them on samples in the On-Manifold ImageNet. However, we would also like to explore the scenario when the given natural images are out-of-manifold. Specifically, we evaluate the trained models by using $\{x_j\}_{j=1}^M$ as normal, and likely out-of-manifold images. Interestingly, from the results presented in Table 4, we observe that all adversarial training methods generalize well for out-of-manifold normal images, while normal training leads to degraded performance. Furthermore, consistent with the findings in previous sections, DMAT leads to comparable robustness with standard adversarial

training against PGD-50 attacks, and an improved robustness compared to standard adversarial training against unseen attacks. These results demonstrate a promising direction of using manifold information (exactly or approximately) to enhance the robustness of deep learning models.

Table 4: Generalization of the defended models to natural (out-of-manifold) images.

| Method | Standard | PGD-50 | Fog | Snow | Elastic | Gabor | JPEG | $L_2$ | $L_1$ |
|---|---|---|---|---|---|---|---|---|---|
| Normal Training | 67.21% | 0.00% | 0.38% | 0.35% | 0.69% | 0.04% | 0.00% | 1.26% | 0.01% |
| AT [PGD-5] | 71.08% | **37.32%** | 23.28% | 45.88% | 46.72% | **46.17%** | 9.00% | 39.72% | 21.21% |
| OM-AT [OM-FGSM] | **79.15%** | 0.19% | 23.41% | 24.00% | 33.53% | 2.53 % | 0.41% | 3.70% | 8.59% |
| OM-AT [OM-PGD-5] | 73.88% | 0.35% | 32.34% | 36.66% | 46.50% | 7.25% | 0.52% | 8.38% | 14.67% |
| DMAT (Ours) | 74.72% | 34.63% | **36.25%** | **50.56%** | **54.14%** | 45.39% | **13.29%** | **48.42%** | **29.68%** |

# 6 Conclusion

In this paper, we investigate whether information about the underlying manifold of natural images can be leveraged to train deep learning models with improved robustness, in particular against unseen and novel attacks. For this purpose, we create the OM-ImageNet dataset, which consists of images with rich textures that are completely on-manifold. The manifold information is exact for OM-ImageNet. Using OM-ImageNet, we show that on-manifold AT cannot defend standard attacks and vice versa. The complimentary benefits of standard and on-manifold AT inspire us to propose dual manifold adversarial training (DMAT) where crafted adversarial examples both in image and latent spaces are used to robustify the underlying model. DMAT improves generalization and robustness to both $L_p$ and non-$L_p$ adversarial attacks. Also, similar improvements are achieved on (out-of-manifold) natural images. These results crystallize the use of the underlying manifold information of images to enhance the intrinsic robustness of the models, specially against novel adversarial attacks.

# 7 Broader Impact

Deep neural networks have been broadly applied to various fields because of their superior performance. However, their vulnerability to adversarial attacks makes general public worry, especially in some security-critical applications such as autonomous vehicle and medical diagnosis. Therefore, we need a method that makes neural networks robust to different attacks. In this work, we explore whether or not leveraging the underlying manifold information could enhance the robustness and generalization capability of neutral networks. We provide an affirmative answer to this question. With the proposed novel OM-ImageNet dataset, future works in this direction could be facilitated. The proposed method DMAT demonstrates a way to enhance both robustness and generalization to known and novel adversarial attacks.

We note that it is possible that future stronger attacks decrease the performance of the proposed defenses. This work should therefore be viewed as a necessary but not sufficient step towards understanding the role of generative image manifolds in the robustness of vision systems against a wide range of adversarial attacks. We anticipate more works on enhancing both robustness and generalization of deep neural networks to gain public confidence in deep learning systems.

## Acknowledgment

This research was supported in part by NSF CAREER AWARD 1942230, HR001119S0026 and Simons Fellowship on "Foundations of Deep Learning" and partially supported by the DARPA GARD Program under the contract DARPA HR001119S0026.

## Footnotes

[3]We use the term "manifold" to refer to the existence of lower-dimensional representations for natural images. This is a commonly used term in the generative model area. However, this definition may not satisfy the exact condition of manifolds defined in mathematics.

[4]There are some perceptible adversarial attacks such as patch attacks that we do not consider in this paper.

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
