[Supplementary Material]

# Supplementary Material:
# Dual Manifold Adversarial Robustness: Defense against $L_p$ and non-$L_p$ Adversarial Attacks

## A    OM-ImageNet Details

### A.1    Overview

In order to construct a dataset consisting of diverse images with exact manifold information, we propose to first train a generator on natural images with a size $256 \times 256$ from 10 super-classes in ImageNet. The natural images are then projected onto the range of the generator, yielding images that lie completely on the manifold defined by the generator. Training generative models for a large-scale natural image dataset (e.g. the complete ImageNet) is known to be a challenge task. Even with the existence of large-scale training of (class-conditional) GANs [1], the diversity of the generated images is still not comparable to the original ImageNet. Therefore, we focus ourselves to a subset of the ImageNet dataset, which is the Mix-10 dataset released by [2]. The original Mix-10 dataset has 77,237 training images and 3,000 test images. To create OM-ImgaeNet with a larger test set, we manually select 69,480 image-label pairs as $\mathcal{D}_{tr}^o = \{x_i, y_i\}_{i=1}^N$ and another 7,200 image-label pairs as $\mathcal{D}_{te}^o = \{x_j, y_j\}_{j=1}^M$. We do not consider the Restricted ImageNet proposed in [3] since Restricted ImageNet has unbalanced classes.

### A.2    StyleGAN training

We use StyleGAN [4] as our generator model, using the default training parameters for $256 \times 256$ images and for four GPUs (summarized in Table 1). We used the training set of the Mix-10 dataset for training. As pre-processing, each image was center-cropped to produce a square image, and converted to $256 \times 256$ resolution.

Table 1: StyleGan Training Parameters

| | |
|---|---|
| Latent space dimensionality | 512 |
| Batch size (at highest resolution) | 16 |
| Training time (total images) | $25 \times 10^6$ |

Formally, the StyleGAN can be written as $G = \tilde{g} \circ h$, where $h : \mathcal{Z} \to \mathcal{W}$ is a mapping network and $\tilde{g} : \mathcal{W} \to \mathcal{X}$ is a synthesis network. We follow [5] and consider the extended latent space of StyleGAN. In [5], it has been shown that embedding images into the extended latent space is easier than $\mathcal{Z}$ or $\mathcal{W}$ space. Therefore, in the following, we consider $g : \mathcal{W}^+ \to \mathcal{X}$ as the generator function which approximates the image manifold With the trained StyleGAN, we project $\mathcal{D}_{tr}^o = \{x_i, y_i\}_{i=1}^N$ and $\mathcal{D}_{te}^o = \{x_j, y_j\}_{j=1}^M$ onto the learned manifold by solving:

$$w_i = \arg\min_w \text{LPIPS}(g(w), x_i) + \|g(w) - x_i\|_1. \tag{1}$$

The resulting on-manifold training and test sets can be represented by: $\mathcal{D}_{tr} = \{(w_i, g(w_i), x_i, y_i)\}_{i=1}^N$, and $\mathcal{D}_{te} = \{(w_j, g(w_j), x_j, y_j)\}_{j=1}^M$. In Figure 1, we present $x_i$ (Original) and $g(w_i)$ (Projected). We can see that the projected images have diverse textures and object

| Original | Projected (On-manifold) | Original | Projected (On-manifold) | Original | Projected (On-manifold) |

Figure 1: Visual comparison between original images and projected images. Even though some fine-grained details are lost after projection, the diversity of the projected images is still high. More importantly, the manifold information for these projected images is *exact*.

Figure 2: Learning rate scheduling during training.

sizes. Moreover, the manifold information for these projected images is exact, which is suitable for investigating the potential benefits of using manifold information in more general scenarios compared to MNIST-like [6, 7, 8] or the CelebA datasets [9].

# B    Implementation Details

## B.1    Classification model training

All the classification models are trained using two P6000 GPUs with a batch size of 64 for 20 epochs. We use the SGD optimizer with the cyclic learning rate scheduling strategy in [10] (see Figure 2), momentum 0.9, and weight decay $5 \times 10^{-4}$.

## B.2    Attack parameters

Formally, given an on-manifold image sample $x = g(w)$, the adversarial perturbation $\delta$ of the standard PGD-$K$ attack can be calculated by:

$$\delta_0 \sim \text{Uniform}[-\epsilon, \epsilon], \quad \delta_{t+1} = \text{Clip}_\epsilon \left[ \delta_t + \epsilon_{iter} \cdot sign(\nabla_{\delta_t} \mathcal{L}(f_\theta(x + \delta_t), y_{\text{true}})) \right], \quad \delta = \delta_K, \quad (2)$$

where $\text{Clip}_\epsilon$ means we clip the perturbation to be within an $L_\infty$ ball $\{\delta : \|\delta\|_\infty < \epsilon\}$. Similarly, the on-manifold PGD-K attack (OM-PGD-K) is given by:

$$\lambda_0 \sim \text{Uniform}[-\eta, \eta], \quad \lambda_{t+1} = \text{Clip}_\eta \left[ \lambda_t + \eta_{iter} \cdot sign(\nabla_{\lambda_t} \mathcal{L}(f_\theta(g(w + \lambda_t)), y_{\text{true}})) \right], \quad \lambda = \lambda_K.$$
$$(3)$$

The parameters for these known attacks are presented in Table 2.

Table 2: Parameter settings for standard and on-manifold attack threat models.

| PGD-5 | PGD-50 | OM-FGSM | OM-PGD-5 | OM-PGD-50 |
|---|---|---|---|---|
| $\epsilon = 4/255$ | $\epsilon = 4/255$ | $\eta = 0.02$ | $\eta = 0.02$ | $\eta = 0.02$ |
| $\epsilon_{iter} = 1/255$ | $\epsilon_{iter} = 1/255$ | $\eta_{iter} = 0.005$ | $\eta_{iter} = 0.005$ | $\eta_{iter} = 0.005$ |

For the unseen attacks proposed in [11], we consider attack parameters presented in Table 3. All these attacks use 200 optimization steps.

Table 3: Parameter settings for the novel attacks.

|  | Fog | Snow | Elastic | Gabor | JPEG | $L_2$ |
|---|---|---|---|---|---|---|
| $\epsilon$ | 128 | 0.062 | 0.500 | 12.500 | 1024 | 1200 |
| Step Size | 0.002 | 0.002 | 0.035 | 0.002 | 72.407 | 170 |

# C   Additional Experiments

## C.1   Effect of the perturbation budgets $\Delta$ and $\Lambda$ in DMAT

In DMAT, $\Delta$ and $\Lambda$ control the strengths of the off-manifold and on-manifold threat models. We study how different choices affect the robustness of the trained networks against unseen attacks. We do not evaluate on known attacks since the performance depends on the type of threat models considered during training.

With the default setting $\epsilon = 4/255, \eta = 0.02$ in the main paper, we manipulate $\epsilon$ (upper-half of Table 4) and $\eta$ (lower-half of Table 4) respectively and evaluate the classification performance. With a stronger off-manifold attack during training, the robustness against unseen attack is higher with the cost of reduced standard accuracy. Interestingly, a stronger on-manifold attack during training leads to both higher standard accuracy and robustness to unseen attacks.

Table 4: Classification accuracy against unseen attacks applied to OM-ImageNet test set.

|  | Standard | Fog | Snow | Elastic | Gabor | JPEG | $L_2$ |
|---|---|---|---|---|---|---|---|
| $\epsilon = 4/255, \epsilon_{iter} = 1/255$ | 77.96% | 31.78% | 51.19% | 56.09% | 51.61% | 14.31% | 51.36% |
| $\epsilon = 2/255, \epsilon_{iter} = 0.5/255$ | 79.29% | 31.91% | 43.45% | 52.82% | 39.15% | 5.84% | 43.67% |
| $\epsilon = 1/255, \epsilon_{iter} = 0.25/255$ | 79.84% | 29.20% | 35.35% | 49.51% | 24.35% | 2.71% | 32.28% |
| $\eta = 0.02, \eta_{iter} = 0.005$ | 77.96% | 31.78% | 51.19% | 56.09% | 51.61% | 14.31% | 51.36% |
| $\eta = 0.01, \eta_{iter} = 0.004$ | 77.34% | 26.35% | 49.49% | 54.07% | 51.63% | 13.22% | 47.81% |
| $\eta = 0.005, \eta_{iter} = 0.002$ | 76.24% | 22.40% | 46.17% | 51.28% | 50.00% | 13.79% | 43.85% |

## C.2   TRADES for DMAT

The proposed DMAT framework is general and can be extended to other adversarial training approaches such as TRADES [12]. In the following, we adopt TRADES in DMAT by considering the following loss function:

$$\min_{\theta} \sum_i \mathcal{L}(f_\theta(x_i), y_{true}) + \beta \max_{\delta} \mathcal{L}(f_\theta(x_i), f_\theta(x_i + \delta)) + \beta \max_{\lambda} \mathcal{L}(f_\theta(x_i), f_\theta(g(w_i + \lambda))), \quad (4)$$

where $x_i = g(w_i)$. The first two terms in (4) are the original TRADES in the image space, and the third term is the counterpart in the latent space. To solve for the two maximization problems in (4), we use PGD-5 and OM-PGD-5 with the same parameter setting in Table 2. Results are presented in Table 5.

Table 5: Classification accuracy against known (PGD-50 and OM-PGD-50) and unseen attacks applied to OM-ImageNet test set. Even for TRADES, the benefit of using manifold information can also be observed.

| Method | PGD-50 | OM-PGD-50 | Fog | Snow | Elastic | Gabor | JPEG | $L_2$ |
|--------|--------|-----------|-----|------|---------|-------|------|-------|
| Normal Training | 0.00% | 0.26% | 0.03% | 0.06% | 1.20% | 0.03% | 0.00% | 1.70% |
| AT [PGD-5] | **38.88%** | 7.23% | 19.76% | 46.39% | 50.32% | 50.43% | 10.23% | 41.98% |
| OM-AT [OM-FGSM] | 0.03% | 20.19% | 11.12% | 13.82% | 34.07% | 1.50% | 0.26% | 2.27% |
| OM-AT [OM-PGD-5] | 0.25% | **27.53%** | 22.39% | 28.38% | 48.74% | 5.19% | 0.49% | 5.92% |
| DMAT | 37.86% | 20.53% | **31.78%** | **51.19%** | 56.09% | 51.61% | 14.31% | 51.36% |
| TRADES | **46.06%** | 8.92% | 18.14% | **47.63%** | 53.32% | **54.33%** | 14.06% | 46.36% |
| DMAT + TRADES | 42.57% | **26.82%** | **30.64%** | 46.62% | **56.38%** | 53.43% | **23.62%** | **55.09%** |

Figure 3: Visual comparison between adversarial training methods when the adversarial examples are crafted using natural (out-of-manifold) images. Brighter colors indicate larger absolute differences. We can observe that the classifier trained with DMAT is more robust and needs stronger distortions to break.

## C.3 Additional Visual Comparisons

We present visual comparisons when the normal images are natural (out-of-manifold) images. Results presented in Figure 3 show that attackers need to apply larger perturbations in order to break the models trained by DMAT.