[Reviews · NeurIPS 2020]

Review 1

Summary and Contributions: The paper investigates the effect on the robustness against different types of adversarial attacks of exploiting information on the true manifold to which the data points belong. In particular, the authors create training and test sets for which the underlying manifold (the term is used in a broad sense) is given by a known StyleGAN from natural images contained in ImageNet. Then, they explore the effect of standard adversarial training, on-manifold adversarial training and a combination of the two, showing that it is possible to achieve at the same time some degree of robustness against standard, on-manifold and even unseen attacks. Finally, similar metrics are tested for out-of-manifold natural images.

Strengths: - The paper generalizes and confirms the findings of previous works about on- and off-manifold robustness to the natural images domain, via introducing the On-Manifold ImageNet dataset, for which the undelying manifold is known. - DMAT, the combination of standard and on-manifold adversarial training, seems novel and shows that manifold information can be useful to achieve robustness to different perturbations. In particular, it outperforms standard adversarial training against unseen attacks - The experimental evaluation is well presented and clearly shows the effects of the different training schemes. - Understanding and disentangling the different types of robustness are important research directions for the field, as well as aiming at models robust against multiple, possibly unseen, adversarial attacks.

Weaknesses: - Methodologically, DMAT is a simple combination of standard and on-manifold adversarial training, already introduced in previous works. The effect of DMAT is a (to some extents predictable) trade-off between the benefits of the two training schemes. - The results are achieved under the strong assumptions of knowing the manifold of the training points. The paper does not discuss how this can be exploited in practice for standard tasks / datasets. In fact, when evaluated on out-of-manifold natural samples (Sec. 5.3) the advantage of DMAT compared to AT on unseen attacks is much less clear, although the natural images are similar to the on-manifold ones (at least looking at the images in the supplementary material).

Correctness: The method and evaluation seem correct, and support the claims.

Clarity: The paper is well-written and present clearly the methods and results.

Relation to Prior Work: The paper discusses quite in details the relation to previous works and the novelty of the proposed method.

Reproducibility: Yes

Additional Feedback: - From Table 2, it appears that the generalization of robustness by adversarial training to unseen attacks is boosted by the addition of OM-AT, although still mainly due AT. How are the results in such metric combining instead AT with other kind of perturbations, e.g. mulitple Lp-norms or Linf-AT+ AT on rotation/translations? Is the improvement due to on-manifold adversarial training or just to more diverse adversarial attacks seen at training time? - Do the authors have an intuition about how the results presented can be useful in practice without the assumption of knowing the exact manifold? As mentioned above, it seems that the benefit of DMAT against unseen attacks decreases with out-of-manifold images. - In [A], a similar approach (using StyleGAN and adversarial training in some latent space) has been used to improve clean accuracy, showing already that manifold information can be useful also in the context of natural images. - Citations [18] and [21] are for the same paper. [A] https://arxiv.org/pdf/1912.03192.pdf ### Update post rebuttal ### I thank the authors for their response. After reading it and the other reviews, I see positively the contribution/experiments on the artificial dataset. In particular, showing the benefit from OM-AT for clean accuracy and OM-robustness also in the domain of natural images is meaningful, combining AT and OM-AT seems also novel, although methodologically quite straightforward and DMAT leads to better robustness against unforeseen attacks (Table 2) on the artificial dataset OM-ImageNet. I'm much less convinced about how the method could be directly applied in practice. The solution suggested in the rebuttal looks a bit weak: in Table 3 AT sometimes achieves better robustness than DMAT on natural images, when these are not used for training (both AT and DMAT use only OM-ImageNet for training, in my understanding). However, in a standard dataset, one would need to compare AT using the whole training set to DMAT using the images projected onto the learnt manifold (according to the pipeline proposed). It is not clear to me if this can be advantageous and would require some extensive experimental evaluation. Also, I agree with R3's comment that Table A is not very meaningful (the attack doesn't consider the projection, the comparison is only with natural training). Overall, I think the merit of the paper still slightly overweighs the weaknesses, hence I keep the original score. I invite the authors to take into consideration all the concerns and suggestions provided in the reviews in the future version of the paper, including those remained unaddressed in the rebuttal, as I think the paper could get significantly strengthened by them.


Review 2

Summary and Contributions: An On-Manifold ImageNet dataset is presented, which is based on a StyleGAN. This allows for reasoning with exactly on-manifold close-to-natural images. The proposed variant of adversarial training DMAT minimizes the empirical worst case losses both for a standard attacker in image space and for an attacker in a threat model based on l-infinity latent space geometry. The result is a model that is quite robust against both these types of attacks as well as several other threat models.

Strengths: The introduction of OM-ImageNet is a solution to the problem that ideas based on natural image data forming a relatively low-dimensional subspace / manifold are normally difficult to evaluate. As those OM projected images look relatively natural, especially in the sense that humans can make out the same semantic features as in the natural originals, and as their robust classifier on OM generalizes to natural, this dataset solution seems valid. This also well enough justifies the theoretical / methodological choices for constructing the dataset. The proposed DMAT scheme combines two known defenses (AT and OM-AT) and achieves robustness against both threat models while retaining the good clean accuracy of OM-AT. Additionally, it shows improved robustness against unforeseen perceptually inspired threat models. Together with the supplement, the paper explains the used method and several design considerations very well.

Weaknesses: A distinction between attacks and threat models should be made. For example, l-infinity AT seems to be quite robust with respects to other l-infinity _attacks_ with the same epsilon ball, but not to attacks of other threat models like with a larger epsilon or like a different geometry than that of l-infinity. Each threat model should be evaluated with a reasonably or even possibly powerful set of attacks, as it is often the case (see the summary Tramer et al. On Adaptive Attacks to Adversarial Example Defenses) that proposed defenses do not hold up robust accuracy in the evaluated threat model when confronted with a more sophisticated attacker. This includes several variations of AT even though the original seems to hold fine. Especially for the standard robustness against image space l-infinity attacks (evaluation listed as PGD-50), many other attacks are known to be successful for different situations and the per-input worst case over them should be reported. While PGD is very good for AT, it alone can not be seen as the strongest attack for evaluation. The visualizations of the different attacks (especially PGD-50 and OM-PGD-50) would be more instructive / comparable if they were shown on the same original. Other strong baselines (like basic TRADES and what is state-of-the-art for the set of unseen attacks) should be included for comparison in the main paper. Especially since generalization to those unseen threat models is a main contribution of the paper, it should be discussed if there are (partially) successful existing approaches and why TRADES does better than AT. From Kang et al. Testing Robustness Against Unforeseen Adversaries, all but the L1-attack are evaluated against. Please include that one too if there is no reason not to.

Correctness: The claims and method look fine, however the robustness claims (especially retained image-space l-infinity robustness) need to be substantiated with a more thorough evaluation.

Clarity: Yes, the text is very clear and comprehensible. (The first supplementary section should be re-read.)

Relation to Prior Work: The paper clearly states the prior work that it is based on, and the proposed dataset and method are to my knowledge novel. For comparison and robustness evaluation, stronger existing approaches should be used.

Reproducibility: Yes

Additional Feedback: Could it happen that the OM attack mostly or partly works as an attack on g, mostly exploiting strange behaviour of that function that f should not have to be robust against? Or is this potential problem already mitigated by the low curvature of g? If it is a concern, would a somehow robustly trained g make sense? Would it be possible to evaluate robustness natural images under an attack similar in threat model to OM-PGD? line 53 supp. The relation between stronger training OM attack and better accuracy can only hold up to a certain point, right? An evaluation on non-adversarial random corruptions from the perceptual categories would be interesting for the supplement. Update: Thanks for the clarifications and especially the additional evaluations in the author feedback. After regarding the other reviews and the rebuttal, I have increased my score from 5 to 6. In my opinion, the paper makes some interesting contributions, especially with the introduction of an on-manifold dataset and the clearly explained proposed methodology. However, it is a borderline decision that I would be fine with going either way, since there are still multiple open questions that the paper would benefit greatly from answering. The evaluation with two additional image space l-infinity attacks that is shown in the rebuttal somewhat solidifies the reported robustness results. Judging from the similarity of DMAT to other methods, it is relatively unlikely that another attack really invalidates the results for standard adversarial robustness; however, evaluations with a more diverse set of attacks should be reported, including some SOTA black box attack. As the other reviewers state as well, the additional results in Table A should only be included if a reasonable end-to-end attack is used, as otherwise they do not give any useful insights about robustness. Since only TRADES+DMAT gets good results on the important unseen attacks when compared, it should be considered to shift some focus of the methodological discussion towards it (maybe more than just moving it from the appendix).


Review 3

Summary and Contributions: Previous work has analyzed on-manifold vs off-manifold adversarial attacks/training. This work attempts to scale up said results to ImageNet by shifting from a VAE-based formulation to a StyleGAN formulation. On real data, is shown that on-manifold adversarial training and standard adversarial training are complimentary. Combining said approaches additionally yields improved performance against unseen perceptual attacks, as well as distinct natural test images, which are out-of-manifold from the perspective of the classifier.

Strengths: The work is easy to read, and the experiments are extensive and meaningful. Demonstrating results which have been obtained on simpler datasets do scale to realistic data is a meaningful contribution in and of itself, and the alternative StyleGAN formulation is algorithmically novel. The observation that on-manifold and standard adversarial training is also novel, providing insight to readers to look further into the difference between said optimization objectives.

Weaknesses: The primary weakness of the paper is the lack of clarity in what is meant by manifold. The process of training StyleGAN and optimizing this heuristic LPIPS + L1 loss will certainly lose information about the underlying data, so though for OM-ImageNet, the underlying manifold is exactly defined, it isn't clear whether it's truly meaningful. Visuals are provided showing the generations are realistic, but there is a gap between quality-looking generations and accurate modeling of the underlying image space. This also makes the evaluation on natural images unclear. Distinct natural test images will be out-of-distribution to a model trained on the StyleGAN latents is clear, but how out-of-distribution isn't clear. The underlying distribution the training and test points were sampled is the same, the Mixed-10 dataset, so the distinction between training manifold and test manifold is entirely artificial. Furthermore, it isn't clear whether finding latents by optimizing said heuristic objective accurately models the training manifold, which makes it even more unclear how these natural images are out-of-distribution. These questions confound the ability to extract insights from the experiments performed, particularly with regards to the natural image experiment.

Correctness: The claims in the paper appear correct, a statement which was striking is that for OM-ImageNet, the underlying manifold information is exact, which can be interpreted as true, however how to interpret it is confounded by the questions on distribution matching and distribution shift raised in the Weaknesses section.

Clarity: The paper is clear to follow, however the core component of the paper, the manifold distinction, is also the component which requires clarification.

Relation to Prior Work: It is clearly discussed and apparent to the reader how this work differs from previous contributions.

Reproducibility: Yes

Additional Feedback: I would suggest the authors provide intuition for the choice of StyleGAN and the optimization objective used to extract latents. I would also suggest the authors to perform experiments quanitfying the distribution mismatch between the training data and the resulting latents via optimization, as well as the distribution mismatch between the resulting latents via optimization and the distinct test data. This will put the experiments in sufficient context to be interpretable. UPDATE: After reading my fellow reviewers' reviews and the rebuttal, I feel this is certainly a borderline paper. R1's concerns are valid and are related to my concerns on the distribution shift with the test set being artificial if the manifold of the training set was truly captured. Table A is not meaningful as the adversary does not have knowledge of the projection. Defining manifold as a low-dimensional data representation is valid, but neither answers the concerns on practicality nor addresses the lack of reasoning for the results on natural examples which the manifold was learned to represent. Though I acknowledge that the contribution is meaningful, given the concerns on practicality and lack of understanding on the distribution shift between the representation and the data after the rebuttal remain, I cannot in good faith say this work is ready for publication. I'd suggest the authors demonstrate DMAT's practicality, where white-box attacks are considered (e.g. adversary is aware of projection), and analyze the distribution shift to provide understanding on why the discrepancies in performance on natural images exists. Why would the data be out-of-distribution to an exact manifold of the data? As with R4, the heuristic choice of objective for learning the manifold could be a source of this. How does the choice of manifold learning objective affect the performance? Can the error in the manifold approximation be related to the observed drop-offs in performance? Analysis is needed before any claims can be made, and this would provide context to the reader on what can be expected out of the given method.


Review 4

Summary and Contributions: This paper deals with adversarial attacks and leverages the empirical success of deep generative models to learn the manifold of images so as to propose Dual Manifold Adversarial Training (DMAT) - a joint adversarial training on a latent space and on the mapped image manifold. A “On Manifold Image Net” dataset is created by projecting a subset of the ImageNet dataset on a latent representation space of a trained StyleGAN. The improvement of the performance on non-adversarial images is discussed, and so is the robustness again attacks in the latent space and in the image space. Additional attacks and “out of manifold” attacks on natural images are studied.

Strengths: Training a model on both latent and image space is an interesting contribution and direction of research. Notably, existing AT methods for robustness in Lp ball may fail against other on manifold attacks and robustness in the latent space may conversely fail against Lp attacks. The empirical evaluation is performed on a reference dataset that provides natural images with significant interest for real-world systems. The possible robustness of this method to unseen attacks (see Table 2) makes it an interesting method.

Weaknesses: The dataset which is proposed relies on manually selected images for which the influence of the selection process upon the experiment results is not well established, and so is the influence of the training method of the StyleGAN, unlike pre-trained models that can be found off-the-shelf. To obtain images that are “completely on-manifold” - which is a strong theoretical statement -, a weighted combination of the LPIPS and a L1 loss is used to define an objective function for an optimization problem for which the solution is a latent representation w of an image x. The existence and uniqueness of such optimization solutions are not discussed so the reader may not understand why the approximate image manifold is exact. This point crucially needs to be clarified. UPDATE: I thank the authors for their clarification on the exact manifold information training.

Correctness: The experimental conclusions drawn from section 4. are promising yet additional details on how the setup was established, the \epsilon = 4/255 radius for the L_{\inf} robustness, the radius \eta = 0.02 on the latent space, the hyperparameter selection for the SGD optimizer and the baselines, the number of epochs, etc. would have been great to strengthen the claims. Could we draw similar experimental conclusions under different configurations? Table 1 presents classification accuracies for different methods and it is interesting to see that DMAT achieves a good trade-off between AT and OM-Ats. It is not clear however how to assess the statistical significance of the figures given. Similarly for Table 2 it is not clear how the figures given are statistically significant, and how performance would change with different possible runs.

Clarity: A well-written review of standard attacks and adversarial robustness strategies is provided, which makes the presentation of the proposed method later in the paper clear and understandable. The structure of the paper however could be improved so as to include more context on how this work relates to prior research for this problem. One hardly understands the relation between the sections at first, transitions between sections and additional descriptions could largely improve the clarity of the work. The notion of manifold is not well established, even though it is used in the literature. However it appears to be at the core of one the key contributions of the paper, namely the OM-ImageNet. Therefore a rigorous explanation is of interest.

Relation to Prior Work: Some prior works are presented in the paper however the relation to it is not well established. Notably, the introduction section motivates the method by presenting some research works. However, no section nor subsection formally positions the paper with regards to the existing works, which makes the novelty of the work as well as the contributions harder to highlight.

Reproducibility: No

Additional Feedback: Could you please clarify how the 68480 images were selected and so for the 7200 image-label pairs? Overall the experiments would be stronger if a clear positioning with regards to prior research would have been made and if more baselines would be provided. Some experiments only look like an ablation study, which still is of great interest. Moreover, the claims would be strengthened by additional datasets for which the setup could be better detailed. UPDATE: specific parameter selection in Sec. 4 (l. 152, 153, 157-159) should again be clarified with regards to the considered partition of the data as well as empirical methodology/significance for future versions or camera ready version of this work.

[Author Response · NeurIPS 2020]

**How DMAT can be exploited for standard tasks/datasets? (R1):** In this work, we consider the scenario when the
manifold information is exact and show that this information can be very useful for improving robustness to novel
attacks. For standard tasks / datasets, one possible pipeline may include the following steps: (1) train a generative model
(e.g. StyleGAN) to capture the approximate manifold (low-dimensional representation) for the dataset, (2) project
the data samples onto the learned manifold, and (3) train a robust classifier using the proposed DMAT. In Table 3 in
the paper, we show that although the classifier is only trained using on-manifold samples, remarkably it demonstrates
good generalization to natural off-manifold samples. To further boost the performance of DMAT for off-manifold
samples, during inference time, one can project the input samples onto the manifold before feeding them to the robust
classifier. In this case, the projection operation is *not* used as a defense mechanism, but as an approach to reduce the
distribution shift between on-manifold samples and natural images. In Table A, we present evaluation results when the
above pipeline is considered. We note that we do not consider end-to-end attacks in this setting since our main focus is
to study the robustness of the classification model itself.

Table A: Evaluation of DMAT on natural images with and without projection against attacks on the classifier.

| Method | Standard | PGD-50 | Fog | Snow | Elastic | Gabor | JPEG | $L_2$ |
|---|---|---|---|---|---|---|---|---|
| Normal Training (ERM) | 67.21% | 0.00% | 0.38% | 0.35% | 0.69% | 0.04% | 0.00% | 1.26% |
| DMAT | 74.72% | 34.63% | 36.25% | 50.56% | 54.14% | 45.39% | 13.29% | 48.42% |
| DMAT + Projection | **77.96%** | **64.39%** | **37.02%** | **65.15%** | **66.47%** | **70.27%** | **72.64%** | **70.77%** |

12
**PGD should not be viewed as the strongest attack for evaluation. (R2):** We agree with the reviewer that considering
a set of adaptive attacks would strengthen our evaluations. Upon your suggestion and for a feasible evaluation runtime,
we now consider FGSM, PGD, and the Momentum Iterative Attacks [1] for the $L_\infty$ threat model. Each test sample will
be mis-classified if one of the attacks fools the classifier (i.e. the per-input worst case). Results are shown in Table B.

$L_1$ **attack should be evaluated. (R2):** Upon your suggestion, we now evaluate our proposed method (DMAT) and
AT model (trained using $L_\infty$) against unseen $L_1$ attacks. Results are presented in the last column of Table B. DMAT
demonstrates improved generalization by around 9 percentage points compared to adversarial training.

Table B: Classification accuracy on OM-ImageNet test set under $L_\infty$ and $L_1$ attacks.

| Method | Standard | FGSM ($L_\infty$) | PGD-50 ($L_\infty$) | MI-PGD-50 ($L_\infty$) | Worst Case ($L_\infty$) | $L_1$ |
|---|---|---|---|---|---|---|
| Normal Training (ERM) | 74.72% | 2.59% | 0.00% | 0.00% | 0.00% | 0.00% |
| AT against $L_\infty$ [PGD-5] | 73.31% | 48.02% | 38.88% | 39.21% | 38.80% | 21.37% |
| DMAT [PGD-5, OM-PGD-5] | 77.96% | 49.12% | 37.86% | 37.65% | 36.66% | 30.70% |

**Other strong baselines such as TRADES should be included in the main paper. (R2):** In the supplementary
material (section C.2), we have presented the results of experiments using TRADES and discussed possible combinations
of DMAT and TRADES. Results show that the generalization ability of TRADES to unseen attacks can also be improved
by exploiting the learned manifold. We will move these results to the main paper and add more discussions on existing
methods for unseen attacks.

**The notion of "manifold" should be clarified. (R3 and R4):** In our paper, manifold refers to the low-dimensional
representation for the data samples. In particular, let the generator $G : \mathbb{R}^r \to \mathbb{R}^d$ where $r \ll d$. The range of the
generator function $G$ is referred to as "manifold". As we indicate in page 2 footnote, this is not the precise definition of
"manifolds" used in topology. We adopted this term since it is commonly used in the generative model area to refer to
the existence of lower-dimensional representations for natural images. We will explain this further in the paper.

**The existence and uniqueness of the optimization solutions are not discussed .... it's unclear why the approx-
imate image manifold is exact. (R4):** For a given natural image $x_i$, we solve for $w_i$ such that $g(w_i)$ is visually
similar to $x_i$ (see some sample results in the supplementary material; Figure 1, On-manifold). The objective we use
is standard and proposed in prior works (e.g. [2]). Since the optimization step is solved by a gradient descent based
method, the solution may not be unique but this is not an issue for training DMAT. We agree that on-manifold samples
$\{g(w)|w \in \mathcal{W}\}$ are approximations to the data samples $\{x_i\}_{i=1}^N$. However, we note that classification model of DMAT,
$\{g(w_i)\}_{i=1}^N$ is used as the training images *not* $\{x_i\}_{i=1}^N$, and therefore the manifold information $g$ for the training set of
DMAT is in fact *exact*. Remarkably, in Table 3 in the main text, we show that the trained classifier has also a very good
generalization to natural images $\{x_j^{test}\}_{j=1}^M$.

**Selection of training images. (R4):** We partition the Mix-10 dataset into 90% training set and 10% test set since the
original test set has a small size. We did not apply any additional curation process in the partition. We will make the
training/test datasets, models and our code publicly available.
[1] Dong *et al.*, "Boosting Adversarial Attacks with Momentum", in CVPR 2018.
[2] Abdal *et al.*, "Image2StyleGAN: How to Embed Images Into the StyleGAN Latent Space?", in ICCV 2019.

[Meta-Review · NeurIPS 2020]

This paper proposes a defense to adversarial examples through generative models and adversarial training. But that's just the pretense of the paper. In reality, the paper is a study of the manifold hypothesis as a defense. The authors construct OM-ImageNet, an ImageNet variant that is entirely on the manifold of a GAN. By doing this, it is possible to evaluate the robustness of defenses that project images onto the manifold of a GAN. Typically it's hard to evaluate manifold projection defenses because images aren't completely on-manifold. This solves that problem. The authors construct a defense for this scheme. I don't believe the claim that this defense works, but the setup of OM-ImageNet is an interesting idea I haven't seen before. Future study on this dataset should be interesting.